

# Structural and functional neuroimaging of hippocampus to study adult neurogenesis in long COVID-19 patients with neuropsychiatric symptoms: a scoping review

Jayakumar Saikarthik[1,2,3], Ilango Saraswathi[4,5], Bijaya Kumar Padhi[6], Muhammad Aaqib Shamim[7], Nasser Alzerwi[8], Abdulaziz Alarifi[9,10] and Aravind P. Gandhi[11]

[1] Department of Maxillofacial Surgery and Diagnostic Sciences, College of Dentistry, Majmaah University, Al Majmaah, Saudi Arabia
[2] SIMATS University, Saveetha Medical College and Hospital, Chennai, Tamil Nadu, India
[3] Department of Medical Education, University of Dundee, Dundee, United Kingdom
[4] Department of Physiology, Madha Medical College and Research Institute, Chennai, India
[5] Department of Business Administration, Manipal Academy of Higher Education, Manipal, India
[6] Department of Community Medicine and School of Public Health, Postgraduate Institute of Medical Education and Research, Chandigarh, India
[7] Department of Pharmacology, All India Institute of Medical Sciences, Jodhpur, India
[8] Department of Surgery, College of Medicine, Majmaah University, Al Majmaah, Saudi Arabia
[9] Department of Basic Sciences, College of Science and Health Professions, King Saud bin Abdulaziz University for Health Sciences, Riyadh, Saudi Arabia
[10] King Abdullah International Medical Research Center, Riyadh, Saudi Arabia
[11] Department of Community Medicine, All India Institute of Medical Sciences, Nagpur, India

Corresponding author
Aravind P. Gandhi,
aravindsocialdoc@gmail.com

## ABSTRACT

**Background.** Worsening of neuropsychiatric and neurodegenerative disorders occurs in COVID-19. Impaired adult neurogenesis is linked to most of the neuropsychiatric symptoms and disorders.

**Aim.** The current scoping review identified and mapped the available evidence on adult neurogenesis in long COVID-19, at a global level following the JBI methodology for scoping reviews and followed the framework by Arksey and O'Malley.

**Method.** Original studies focusing on structural and functional neuroimaging of the hippocampus to study adult neurogenesis in long COVID-19 were included in the review. Studies published in English language with no restriction on the time of publication were searched using the specified search strategy in PubMed, Web of Science, Embase, and SCOPUS. Articles obtained from the database search were collated and uploaded into the Nested Knowledge AutoLit semi-automated systematic review platform for data extraction.

**Results.** The current review provides evidence of the potential alterations in adult neurogenesis in long COVID-19 and its potential link to neuropsychiatric sequelae of long COVID-19, with further research required to validate this assertion.

**Conclusion.** This review proposes conceptual and methodological approaches for future investigations to address existing limitations and elucidate the precise role of adult neurogenesis in the pathophysiology and treatment of long COVID-19.

## INTRODUCTION

The presence of persistent symptoms beyond the acute phase of coronavirus diseases, as seen in SARS and MERS outbreaks, echoes in the ongoing COVID-19 pandemic (*Ahmed et al., 2020*; *Hui et al., 2005*). While the acute phase of COVID-19 is typically considered to span three to four weeks from symptom onset, symptoms persisting beyond this period constitute what is known as long COVID-19 (LC) or post-acute sequelae of COVID-19 (PASC) (*Nalbandian et al., 2021*). Long COVID-19 encompasses a spectrum of debilitating symptoms affecting multiple organ systems which may manifest as various syndromes, including post-intensive care unit syndrome, post-viral fatigue syndrome, long-term COVID syndrome, and irreversible organ damage (*Zavaleta-Monestel et al., 2024*). Patients with post-acute COVID-19 syndrome often experience a range of neuropsychiatric symptoms, such as anosmia, ageusia, sleep disturbances, cognitive impairments, and mood disorders (*Badenoch et al., 2022*). Moreover, exacerbations of existing neuropsychiatric conditions like Alzheimer's disease, Parkinson's disease, major depressive disorder, anxiety, schizophrenia, and dementia have been observed in COVID-19 patients (*Cilia et al., 2020*; *Gan et al., 2021*; *Gobbi et al., 2020*; *Pourfridoni & Askarpour, 2023*).

Adult neurogenesis (AN), the development of new neurons from neural stem cells in the adult brain, has been a subject of intense scientific inquiry. Initially dismissed, evidence from the late 20th and early 21st centuries have shifted the consensus towards acknowledging adult neurogenesis, albeit in specific brain regions and under certain conditions (*Abrous, Koehl & Le Moal, 2005*; *Altman & Das, 1965*; *Gould & Gross, 2002*; *Moreno-Jiménez et al., 2021*; *Owji & Shoja, 2020*). Its functions span learning, memory, emotions, olfaction, stress response, behavior, brain repair following injury, and brain plasticity (*Apple, Fonseca & Kokovay, 2017*). Dysregulation of adult neurogenesis is associated with various neurodegenerative and neuropsychiatric disorders, including Parkinson's disease, depression, anxiety, dementia, schizophrenia, and Alzheimer's disease (*Cho et al., 2015*; *Hussaini et al., 2014*; *Moreno-Jiménez et al., 2021*).

Given the speculation surrounding the impact of COVID-19 on adult neurogenesis (*Kumaria, Noah & Kirkman, 2022*; *Saikarthik et al., 2022*), understanding the potential link between long COVID-19 and AN prompts three critical questions. First, which brain areas are involved in adult neurogenesis? The subventricular zone (SVZ) of the lateral ventricles and the subgranular zone (SGZ) of the dentate gyrus of the hippocampus are major sites of adult neurogenesis (*Feliciano, Bordey & Bonfanti, 2015*).

Second, is there evidence of COVID-19 affecting these brain areas? Emerging evidence suggests that the hippocampus is notably susceptible to hypoxia and hypo-perfusion, which shows higher alterations compared to other brain regions, specifically in the cornu ammonis and dentate gyrus subfields (*Fotuhi, Do & Jack, 2012*). There is a potential scenario where certain individuals might lack a sufficient number of multipotent progenitors in the

neurogenic niche for neurogenesis to restart after an insult (*Boldrini et al., 2018*). Treatment of human hippocampal progenitor cells with serum from acute COVID-19 patients with delirium reduced cellular proliferation and neurogenesis while increasing apoptosis, compared to serum from age- and sex-matched COVID-19 patients without delirium. These effects were mediated by IL-6, which triggered downstream cytokines IL-12 and IL-13 (*Borsini et al., 2022*). Long COVID-19 patients with persistent cognitive symptoms showed elevated levels of CCL11—a chemokine known to activate hippocampal microglia and impair neurogenesis—compared to those without such symptoms. Similarly, mice with mild COVID-19 exhibited sustained hippocampal neurogenesis impairment, reduced oligodendrocytes, myelin loss, and increased CSF cytokine levels (*Fernández-Castañeda et al., 2022*). Both COVID-19 hamsters and humans that died from COVID-19 showed fewer neuroblasts and immature neurons in the dentate gyrus (*Soung et al., 2022*). SARS-CoV-2 increases brain IL-1$\beta$, leading to IL-1R1-mediated loss of hippocampal neurogenesis and post acute cognitive deficits in mice (*Vanderheiden et al., 2024*). Postmortem hippocampal samples from COVID-19 patients showed neuronal apoptosis, reduced neurogenesis, altered pyramidal cell morphology, and disrupted astrocyte and microglia distribution (*Bayat et al., 2022*). Mendelian randomization studies and longitudinal investigations confirm a causal link between severe COVID-19 and reduced hippocampal volume (*Zhou et al., 2023*). Notably, persistent structural, functional, and cognitive changes, including hippocampal volume reduction, are observed in mild COVID-19 cases even one year post-infection (*Invernizzi et al., 2023*). Additionally, COVID-19 patients without neurological symptoms display hippocampal grey matter atrophy and reduced blood flow (*Qin et al., 2021*). Large-scale studies corroborate these findings, indicating significant grey matter loss in the hippocampus of COVID-19 patients compared to controls (*Douaud et al., 2022*; *Ma et al., 2022*; *Shan et al., 2022*). These findings underscore the intricate relationship between viral infection, inflammation, and hippocampal alterations in acute and long COVID-19.

Third, how can adult neurogenesis alterations in long COVID-19 patients be detected? Tracking and quantifying new neurons in the adult brain can be technically demanding and subject to various sources of error. Measuring adult neurogenesis in live humans is a challenging task because it involves tracking the birth and maturation of new neurons in the brain, which is not easily observable through standard imaging techniques. True measurement of adult neurogenesis can be done reliably only on post-mortem samples. However, researchers have developed several indirect and non-invasive methods to study adult neurogenesis in humans including neuroimaging techniques like magnetic resonance imaging (MRI), functional MRI (fMRI), positron emission tomography (PET). While MRI cannot directly visualize neurogenesis, it can be used to measure changes in the volume of brain regions associated with neurogenesis. An increase in hippocampal volume over time may suggest the presence of ongoing neurogenesis (*Horgusluoglu-Moloch et al., 2019*; *Killgore, Olson & Weber, 2013*). Functional MRI can be used to assess the functional connectivity and activity of the hippocampus, which can indirectly reflect changes associated with neurogenesis (*Burdette et al., 2010*; *Cheng et al., 2019*; *Yassa et al., 2011*). Some PET tracers, such as [18F] FLT, have been used to estimate cell proliferation in the brain (*Rueger et al., 2010*). However, these methods are still experimental and have

limitations. The methods mentioned above are often indirect and may provide only an estimation of neurogenesis. Additionally, the results can be influenced by various factors, including age, genetics, and environmental factors. Unwinding of AN induced variations in the neuroimaging findings from those by other confounders is still a challenge that is acknowledged and necessarily overlooked occasionally by the scientific community in studying AN in live humans (*Just, Chevillard & Migaud, 2022*). The readers are urged to refer to reviews by *Ho et al. (2013)* and *Just, Chevillard & Migaud (2022)* for further clarification on *in vivo* imaging of adult neurogenesis in humans.

A preliminary search of MEDLINE, the Cochrane Database of Systematic Reviews, and JBI Evidence Synthesis revealed no current or ongoing systematic or scoping reviews specifically addressing adult neurogenesis in long COVID-19. This gap in the literature highlights the novelty of this scoping review, which aims to map the available evidence on this topic. By synthesizing existing research, the review seeks to enhance understanding of the neurological impacts of long COVID-19, particularly in relation to neurogenesis, and to identify areas requiring further investigation. The findings will inform healthcare professionals, guide policymakers in devising effective strategies, and support future research in this emerging field.

## MATERIALS AND METHODS

Preferred Reporting Items for Systematic Reviews and Meta-Analysis Protocols (PRISMA-P) was used to prepare the protocol and PRISMA-ScR extension for scoping reviews was used to present the results (*Tricco et al., 2018*; *Moher et al., 2015*). The current scoping review was conducted following the guidelines framed by *Arksey & O'Malley (2005)* employing the following steps: identifying the research question, identifying relevant studies, selection of eligible studies, charting the data, and collating, summarizing and reporting of results.

### Identifying the research question

The principal research question was "What is the current evidence on the impact of long COVID/PASC on adult neurogenesis and its association with neuropsychiatric symptoms?"

The principal research question was broken down into the following:

I   What type of studies have been conducted on adult neurogenesis in patients with long COVID-19/PASC?

II   What neuropsychiatric symptoms are commonly reported in patients with long COVID-19/PASC?

III   What are the proposed mechanisms linking long COVID-19/PASC to changes in adult neurogenesis?

IV   How are changes in adult neurogenesis measured in patients with long COVID-19/PASC?

V   What interventions or treatments have been studied to address changes in adult neurogenesis in patients with long COVID-19/PASC?

VI   What are the gaps that exist in the current literature on this topic?

### Identifying relevant studies

The scoping review followed the Population, Concept, Context (PCC) format to identify relevant studies for answering the research questions with population being patients diagnosed with long COVID-19 with neuropsychiatric sequalae, and concept being adult neurogenesis.

The current scoping review was conducted in accordance with the JBI methodology for scoping reviews (*Peters et al., 2020*). A preliminary limited search was conducted in PubMed to identify articles published on the topic. The keywords and Mesh terms from the identified topics were used to develop the search strategy for PubMed, Web of Science, Embase, and SCOPUS. Studies published in the English language with no restriction on the time of publication were searched using the keywords "Neurogenesis [Mesh]", "Adult neurogenesis[tiab]", "Post-acute COVID-19 Syndrome [Mesh]", long COVID-19[tiab], and SARS-CoV-2[tiab] using the search strategy as given in Table S1.

This scoping review included a range of study designs, incorporating both experimental and quasi-experimental approaches such as randomized controlled trials, non-randomized controlled trials, before-and-after studies, and interrupted time-series studies. Additionally, it encompassed analytical observational studies including prospective and retrospective cohort studies, case-control studies, and analytical cross-sectional studies. Descriptive observational study designs such as case series, individual case reports, and descriptive cross-sectional studies, were also considered for inclusion. However, opinion articles, as well as systematic reviews were excluded from this scoping review.

### Selection of eligible studies

All identified articles obtained from the database search were collated and uploaded to Nested Knowledge AutoLit semi-automated systematic review platform and duplicates were removed. Following a pilot test, two reviewers (IS & JS) independently screened the titles and abstracts against the inclusion criteria for the scoping review.

Only original research articles conducted on human subjects with focusing on adult neurogenesis in long COVID-19 patients were included in the review. While review articles, opinion papers, and studies where full-text articles could not be retrieved were excluded. Full texts of potentially relevant sources were retrieved, and their citation details were imported into the JBI System for the Unified Management, Assessment, and Review of Information (JBI SUMARI) (JBI, Adelaide, Australia) (*Munn et al., 2019*). Two independent reviewers (IS & JS) thoroughly evaluated the complete texts of chosen articles against the specified eligibility criteria. The studies that did not meet these criteria were documented and included in the scoping review report as excluded evidence. Disagreements among the reviewers during the selection process were resolved through discussion.

### Charting the data

A draft version of data extraction tool was created and the preliminary data extraction tool underwent adjustments and refinements as needed throughout the data extraction process for each included source of evidence. These modifications were documented within the scoping review. Any discrepancies between reviewers were addressed through discussion.

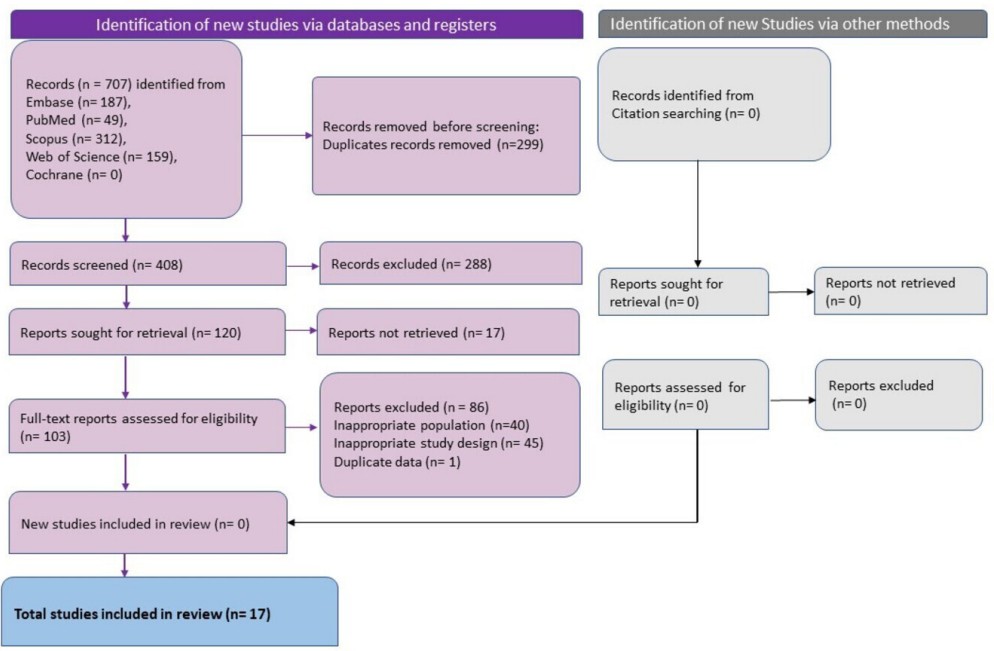

**Figure 1** The summary of the data search and extraction process done using the nested knowledge tool.

## Collating, summarising, and reporting the results

The extracted data were summarised and reported as tables (Tables 1 and 2). A narrative format is used to describe the results in relation to the review questions and study purpose.

## RESULTS

### Overview of results

Database search revealed 707 studies and after removal of duplicates 408 unique studies remained. Following title and abstract screening, 288 studies were excluded, and 120 studies were sought for full-text retrieval and screening. 103 studies were excluded after full-text screening and 17 studies published between 2020 to 2023 which adhered to the eligibility criteria were included in this review (Fig. 1) (*Balsak et al., 2023*; *Barnden et al., 2023*; *Besteher et al., 2022*; *Carroll et al., 2020*; *Cattarinussi et al., 2022*; *Díez-Cirarda et al., 2023*; *Du et al., 2022*; *Ergül et al., 2022*; *Esposito et al., 2022*; *Franke et al., 2023*; *Lu et al., 2020*; *Muccioli et al., 2023*; *Rothstein, 2023*; *Taskiran-Sag et al., 2023*; *Tu et al., 2021*; *Wingrove et al., 2023*; *Yousefi-Koma et al., 2021*). Three studies each were conducted in Italy (*Cattarinussi et al., 2022*; *Esposito et al., 2022*; *Muccioli et al., 2023*), China (*Du et al., 2022*; *Lu et al., 2020*; *Tu et al., 2021*), and Turkey (*Balsak et al., 2023*; *Ergül et al., 2022*; *Taskiran-Sag et al., 2023*). Two studies were conducted in Germany (*Besteher et al., 2022*; *Franke et al., 2023*) and the United States of America (*Carroll et al., 2020*; *Rothstein, 2023*), and one study was conducted in Spain (*Díez-Cirarda et al., 2023*), Iran (*Yousefi-Koma et al.,*

**Table 1 *In vivo* correlates to detect AN.**

| Imaging techniques | *In vivo* neural correlates of adult neurogenesis |
| --- | --- |
| Voxel-based morphometry (VBM) | Increased hippocampal grey matter volume and hippocampal volume denotes enhanced AN (*Han et al., 2020*; *Horgusluoglu-Moloch et al., 2019*; *Killgore, Olson & Weber, 2013*) |
| Blood oxygen level dependent (BOLD) functional magnetic resonance imaging (fMRI) | Increased BOLD fMRI signals in dentate gyrus/CA3 region denotes enhanced AN (*Yassa et al., 2011*) Increased functional connectivity with other brain regions and within hippocampus denotes enhanced AN (*Burdette et al., 2010*) Decreased RoHo and decreased ALFF in hippocampus denotes impaired AN (*Cheng et al., 2019*). |
| MRI cerebral-blood-volume (CBV) | Increased cerebral blood volume within dentate gyrus, hippocampus denotes enhanced AN (*Pereira et al., 2007*) |
| MRI cerebral-blood-flow (CBF) | Increased cerebral blood flow within dentate gyrus, hippocampus denotes enhanced AN (*Steventon et al., 2020*) |
| Diffusion-weighted and diffusion tensor imaging (DWI/DTI) | Increased mean diffusivity (MD) and decrease in fractional anisotropy (FA) in SVZ denotes impaired adult neurogenesis (*Cherubini et al., 2010*). |
| Magnetic resonance spectroscopy (MRS) | Biomarker peaking at 1.28 ppm that arise from lipids during AN denotes enhanced AN (*Manganas et al., 2007*). |

**Notes.**
Abbreviation: AN, Adult neurogenesis; SVZ, Subventricular zone; ALFF, amplitude of low-frequency fluctuations; MD, Mean diffusivity; FA, fractional anisotropy.

*2021*), UK (*Wingrove et al., 2023*), and Australia (*Barnden et al., 2023*). The studies varied in sample size with the median being 49 patients. The studies with the smallest sample size were case reports with one sample (*Carroll et al., 2020*; *Yousefi-Koma et al., 2021*) and the largest sample size was 252 (126 each in the study group and control group) (*Tu et al., 2021*). The total number of patients studied in all the studies put together was 1,114 with 687 in the study group and 427 in the control group. Gender distribution varied between studies. The case reports were on females ($n = 1$) (*Carroll et al., 2020*; *Yousefi-Koma et al., 2021*) and the remaining studies included both male and female participants. The median of the percentage of male participants across the studies was 35.6%. The age of the participants ranged between 28 and 69 years with the mean ($\pm$SD) age of 44.07 $\pm$ 8.96 years (Table 3).

## Criteria for selecting long COVID-19 patients

There were minor disparities in the criteria for defining long COVID-19 by the selected studies. Most studies defined long COVID (LC) using WHO or NICE guidelines (*Barnden et al., 2023*; *Franke et al., 2023*; *Muccioli et al., 2023*; *Rothstein, 2023*), while others included participants based on persistent symptoms beyond 12 weeks (*Díez-Cirarda et al., 2023*; *Wingrove et al., 2023*) or broader post-acute symptom duration (*Balsak et al., 2023*; *Besteher et al., 2022*; *Cattarinussi et al., 2022*; *Du et al., 2022*; *Ergül et al., 2022*; *Esposito et al., 2022*; *Lu et al., 2020*; *Taskiran-Sag et al., 2023*; *Tu et al., 2021*; *Yousefi-Koma et al., 2021*) WHO defines long COVID-19 as "the continuation or development of new symptoms 3 months

**Table 2   Summary of hippocampal pathology in long COVID-19.**

| Major finding | Volumetric changes | Functional connectivity changes | Perfusion changes | Biomarker changes | Supporting evidence/observation | Reported mechanism |
|---|---|---|---|---|---|---|
| Volumetric Change - Yes | *Besteher et al. (2022)* | – | – | – | Larger bilateral hippocampal GMV in LC (*Besteher et al., 2022*) | 1. Recovery due to AN and/or increased functional activity leading to hypertrophy of neurons and amplications of dendritic connections. 2. Neuroinflammation |
| | *Lu et al. (2020)* | | | | Higher bilateral hippocampal grey matter volume (GMV) in LC compared to healthy controls | Neurogenesis and functional compensation |
| | *Tu et al. (2021)* | | | | Higher GMV in bilateral hippocampus in LC than HC. | Functional compensation to cope with the acute stress and the ongoing COVID-19 related trauma |
| | *Rothstein (2023)* | | | | Larger hippocampal volume compared to computerised normalised volumes | Macroscopic volume changes associated with adult hippocampal neurogenesis |
| Volumetric Change - No | *Taskiran-Sag et al. (2023)* | – | – | – | No difference in bilateral hippocampal thickness between LC and HC | – |
| | *Ergül et al. (2022)* | | | | No difference in bilateral hippocampal volume between LC and HC | – |
| | *Yousefi-Koma et al. (2021)* | | | | Normal hippocampal volume denoting functional impairment, not structural | Neuroinvasion by COVID-19 |
| Volumetric Change - Conflicting | *Díez-Cirarda et al. (2023)* | – | – | – | Lower volume in all hippocampal subfields in LC when compared to HC except in CA3 body and parasubiculum subfields | 1. Acute damage, such as hypoxia or acute neuroinflammation 2. Consequence of persistent neuroinflammation and compensatory mechanism driven by astrocyte activation, and reduction of neurogenesis inhibition in the hippocampus 3. Unchaining of neurodegenerative mechanisms. |
| | *Muccioli et al. (2023)* | | | | Bilateral hippocampal volume reduction in LC when compared to HC but the statistical significance was lost when corrected for multiple comparison | Loss of sensory input due to anosmia, neuroinflammatory events, or neurodegeneration |

Peer J

**Table 2** (*continued*)

| Major finding | Volumetric changes | Functional connectivity changes | Perfusion changes | Biomarker changes | Supporting evidence/observation | Reported mechanism |
|---|---|---|---|---|---|---|
| Functional Connectivity - Yes | – | *Cattarinussi et al. (2022)* | – | – | Increase in intrinsic functional connectivity in the right hippocampus | COVID-19 causes changes in local FC in areas primarily involved in social behaviour and mood regulation, leading to the development of depressive symptoms, possibly mediated by an inflammatory response. |
| | | *Esposito et al. (2022)* | | | Reduction in functional connectivity between hippocampus and insula in LC with olfactory disorder | The olfactory network functions similarly to a cognitive reserve network, and olfactory loss may serve as a sensory indicator of reduced neural plasticity, a characteristic reserve affected by COVID-19. |
| | | *Barnden et al. (2023)* | | | Weaker connectivity between bilateral hippocampus and the whole brain in LC when compared to HC | Impaired cell membrane calcium transport and transient receptor potential melastatin 3 (TRPM3) reported dysfunction in LC |
| | | *Díez-Cirarda et al. (2023)* | | | Reduced functional connectivity in PCS patients compared HC between the right head of the hippocampus and the left anterior Para hippocampal division and the parietal area, including supra-marginal and postcentral areas, overlapping the dorsal attention network | 1. Acute damage, such as hypoxia or acute neuro-inflammation 2. Consequence of persistent neuroinflammation and compensatory mechanism driven by astrocyte activation, and reduction of neurogenesis inhibition in the hippocampus 3. Unchaining of neurodegenerative mechanisms. |
| Functional Connectivity - No/Conflicting | – | *Muccioli et al. (2023)* | – | – | 1. No differences in functional connectivity of hippocampus between LC and HC. 2. The olfactory network of patients with COVID-19-related olfactory dysfunction was overall less segregated into clusters of functionally associated components, which likely play specific functions in the central olfactory processing. 3. Dysfunctional connectivity between right thalamus and right posterior hippocampus | Loss of sensory input due to anosmia, neuroinflammatory events, or neurodegeneration |
| | | *Franke et al. (2023)* | | | Cranial MR imaging did not reveal pathological findings correlating with cognitive impairment including atrophy. | |

**Table 2** (*continued*)

| Major finding | Volumetric changes | Functional connectivity changes | Perfusion changes | Biomarker changes | Supporting evidence/observation | Reported mechanism |
|---|---|---|---|---|---|---|
| Perfusion Change - Yes | – | – | *Díez-Cirarda et al. (2023)* | – | Lower hippocampal perfusion in LC compared to HC | 1. Acute damage, such as hypoxia or acute neuro-inflammation 2. Consequence of persistent neuroinflammation and compensatory mechanism driven by astrocyte activation, and reduction of neurogenesis inhibition in the hippocampus 3. Unchaining of neurodegenerative mechanisms. |
| | | | *Wingrove et al. (2023)* | | 1.Higher CBF in posterior hippocampus in LC with anosmia when compared to those COVID-19 patients with resolved anosmia. 2. No changes in global GM perfusion and no difference in CBF between COVID patients and HC | 1. Subtle vascular effects that are only noticeable in the smaller diameter arteries supplying the brain (posterior cerebral artery) 2. Local changes in neuronal activity/metabolism, due to the generally accepted coupling between blood flow and metabolism |
| Perfusion Change - No | – | – | *Franke et al. (2023)* | – | Cranial MR imaging did not reveal pathological findings correlating with cognitive impairment including atrophy. | Humoral autoimmunity may contribute to the development of cognitive impairment in some PASC patients |
| Biomarker Change - Yes | – | – | – | *Díez-Cirarda et al. (2023)* | 1. Increased GFAP, MOG and Nfl in LC when compared to HC. 2. Positive correlation between GFAP and MOG and whole hippocampal volume 3. CCL11 (inhibitor of AN) showed negative and significant associations with dentate gyrus, CA3 head and CA4 head volumes of the hippocampus and NfL showed negative and significant associations with hippocampal head subfield | 1. Acute damage, such as hypoxia or acute neuro-inflammation 2. Consequence of persistent neuroinflammation and compensatory mechanism driven by astrocyte activation, and reduction of neurogenesis inhibition in the hippocampus 3. Unchaining of neurodegenerative mechanisms. |
| | | | | *Carroll et al. (2020)* | Elevated systemic inflammatory markers, CRP | |
| | | | | *Balsak et al. (2023)* | Negative correlation between FA from hippocampus and plasma LDH levels in LC patients who were hospitalized | Deterioration of axonal integrity and demyelination process secondary to hypoxia in hippocampus. |
| Biomarker Change - No | – | – | – | *Taskiran-Sag et al. (2023)* | No correlation between hippocampal thickness and inflammatory biomarkers CRP and neutrophil to lymphocyte ratio | |
| | | | | *Ergül et al. (2022)* | No correlation between hippocampal volume and biochemical parameters vitamin B12, Zn, Fe, ferritin, T4, TSH and endo-cannabinoids | |
| | | | | *Du et al. (2022)* | No correlation between ALFF in hippocampus and inflammatory biomarkers CRP, neutrophil and lymphocyte count | Compensatory repair of brain tissue following hypoxia or inflammation. |

**Notes.**

Abbreviation: LC, Long COVID-19; GMV, Gray matter volume; HC, healthy controls; FC, functional connectivity; AN, adult neurogenesis; NPS, neuropsychiatric symptoms; ALFF, amplitude of low-frequency fluctuations; MD, Mean diffusivity; FA, fractional anisotropy; AD, Axial diffusivity; RD, radial diffusivity; CRP- C, reactive protein; TSH, thyroid stimulating hormone; LDH, Lactate dehydrogenase; GFAP, Glial Fibrillary Acidic Protein; MOG, Myelin Oligodendrocyte Glycoprotein; NfL, Neurofilament Light Chain; CCL11, Eotaxin-1; CBF, cerebral blood flow; GM, Gray matter; PCS, Post COVID-19 syndrome.

**Table 3  Study characteristics.**

| Author., et al. (Year) | Country | Study population/cluster | Long COVID-19 definition | Sample size | | Age | | Gender (Male) | |
|---|---|---|---|---|---|---|---|---|---|
| | | | | Study group | Control group | Study group | Control group | Study | Control |
| | | | | | | Mean ± SD | Mean ± SD | N (%) | N (%) |
| Besteher et al. (2022) | Germany | Patients from the post-COVID out-patient clinic of the Department of Internal Medicine (Infectiology) and the Department of Neurology of Jena University Hospital in Germany | Not reported | 30 LC with NPS | 20 | 47.5 ± 11.5 | 42.95 ± 13.41 | 13 (43.3%) | 10 (50%) |
| Cattarinussi et al. (2022) | Italy | Patients enrolled at the University Hospital of Padova, Padua, Italy from May to November 2020. | Not reported | 79 (44 LC) | 17 | 42.8 ± 13.8 | 35.8 ± 11.7 | 33 (41.7%) | 11 (57.9%) |
| Du et al. (2022) | China | Patients who had been admitted with COVID-19 and discharged from the First Hospital of Changsha about 1 year earlier. | Not reported | 19 LC | 25 | 54.21 ± 8.7 | 50.48 ± 11.58 | 8 (42.1%) | 7 (28%) |
| Lu et al. (2020) | China | Recovered COVID-19 patients who were discharged from Fuyang No.2 People's Hospital | Not reported | 60 | 39 | 44.1 ± 16 | 45.88 ± 13.9 | 34 (56.67%) | 22 (56.41%) |
| Taskiran-Sag et al. (2023) | Turkey | Patients who were assessed in the out-patient clinics, Department of neurology, Ankara, Turkey, with a confirmed COVID-19 medical history. | Not reported | 20 | 20 | 35.5 ± 9.5 | 36.3 ± 6.7 | 10 (50%) | 11(55%) |
| Ergül et al. (2022) | Turkey | COVID-19 patients who were diagnosed with COVID-19 with a PCR test in the Otorhinolaryngology Outpatient Clinic of the Kastamonu Research and Training Hospital. | Not reported | 20 | 20 | 34.25 ± 13.05 | 32.2 ± 9.9 | 12 (60%) | 11(55%) |
| Esposito et al. (2022) | Italy | Subjects with previous SARS-CoV-2 infection from the geographical area of Naples (Italy) | Not reported | 27 | 18 | 40 ± 7.6 | 36 ± 7.1 | 10 (37%) | 6 (33.3%) |
| Yousefi-Koma et al. (2021) | Iran | | Not reported | 1 | 0 | 28 | | 1 F | |
| Rothstein (2023) | USA | COVID-19 patients with long term neurologic symptoms. | WHO and NICE guidelines | 24 | 0 | 46.9 (Range 22–60) | | 5 (21%) | |
| Balsak et al. (2023) | Turkey | COVID-19 patients | Not reported | 74 | 52 | 42.92 ± 18.01 | 41.62 ± 12.17 | 46 (51.7%) | 25 (48.1%) |
| Tu et al. (2021) | China | COVID-19 survivors with a clinical diagnosis and were discharged from hospitals in Wuhan, China | Not reported | 126 | 126 | 52.4 ± 13.5 | 52 ± 13.3 | 40 (31.7%) | 40 (31.7%) |
| Díez-Cirarda et al. (2023) | Spain | Patients who attended the department of Neurology at Hospital Clínico San-Carlos between November 2020 and December 2021 | Patients with a history of SARS-CoV-2 infection with persistent symptoms over 12 weeks are diagnosed with post-COVID syndrome | 84 | 33 | 50.89 ± 11.25 | 49.18 ± 16.14 | 26 (31.96%) | 13 (39.4%) |
| Wingrove et al. (2023) | UK | People who were not hospitalised and were vaccine naïve and who still had impaired olfactory function 4–6 weeks after initial COVID-19 infection | Persistent symptoms following COVID-19 infection beyond 12 weeks is defined as 'long COVID-19 | 39 (8 LC) | 18 | 52.25 ± 12.17 | 38.89 ± 11.39 | 9 (23.1%) | 9 (50%) |
| Franke et al. (2023) | Germany | COVID-19 patients with residual neurological symptoms who attended two German university hospitals with specialized neurology outpatient clinics | WHO-PCS is defined by new or ongoing symptoms three months after the onset of acute COVID-19 that last for at least 2 months, fluctuate in appearance, and are not explained by another diagnosis | 50 | 0 | 46.92 ± 11.68 | | 17 (34%) | |
| Muccioli et al. (2023) | Italy | Patients who attended IRCCS Istituto delle Scienze Neurologiche di Bologna, Bologna, Italy) | WHO and NICE guidelines | 23 | 26 | 37 ± 14 | 38.5 ± 13.7 | 11 (47.8%) | 13 (50%) |
| Carroll et al. (2020) | USA | | Not reported | 1 | 0 | 69 | | 1 f | |
| Barnden et al. (2023) | Australia | fatigue affected LC patients | WHO working case definition | 10 LC | 13 | 44 ± 15 | 39 ± 13 | 3 (33.3%) | 6 (46.15%) |

**Notes.**

Abbreviations: LC, Long COVID-19; PCS, Post COVID-19 syndrome; NPS, Neuropsychiatric symptoms; NICE, National Institute for Health and Care Excellence; SD, standard deviation.

after the initial SARS-CoV-2 infection, with the symptoms lasting for at least 2 months with no other explanation" (*WHO, 2022*). NICE guidelines give two definitions of post-acute COVID-19 including "(1) ongoing symptomatic COVID-19 for people who still have symptoms between 4 and 12 weeks after the start of acute symptoms; and (2) post-COVID-19 syndrome for people who still have symptoms for more than 12 weeks after the start of acute symptoms" (*Venkatesan, 2021*). Taking into consideration the participant characteristics, symptomology, and day of the scan/study post-infection, those studies conducted on COVID-19 patients with persistent symptoms for a minimum of 4 weeks were also included in the review even though the authors did not exclusively mention the study to be conducted on long COVID-19 patients (*Balsak et al., 2023*; *Carroll et al., 2020*; *Cattarinussi et al., 2022*; *Du et al., 2022*; *Ergül et al., 2022*; *Esposito et al., 2022*; *Lu et al., 2020*; *Taskiran-Sag et al., 2023*; *Tu et al., 2021*; *Yousefi-Koma et al., 2021*) (Table 3). The common challenge faced by the authors is linking the complaints to COVID-19 after recovery from the disease and considering the patients to be COVID-19 long haulers. Using matching controls was one of the ways employed by the authors to overcome this challenge. Except the study by *Franke et al. (2023)* and *Rothstein (2023)* which was conducted on a cohort of long COVID-19 patients with self-reported cognitive deficits and in COVID-19 patients with long term neurologic symptoms, the remaining research articles were case control studies (*Balsak et al., 2023*; *Cattarinussi et al., 2022*; *Du et al., 2022*; *Ergül et al., 2022*; *Esposito et al., 2022*; *Lu et al., 2020*; *Taskiran-Sag et al., 2023*; *Tu et al., 2021*). The day of the brain scan post infection across the studies are given in Table S2.

## Clinical type of COVID-19 in the included studies

Majority of the studies were conducted in non-critical COVID-19 patients though the severity of COVID-19 experienced by the study participants were not mentioned in a few studies (34.06% of study group) (*Cattarinussi et al., 2022*; *Díez-Cirarda et al., 2023*; *Ergül et al., 2022*; *Franke et al., 2023*; *Yousefi-Koma et al., 2021*). In the participants for whom the COVID-19 severity was reported, 78.14% were of the mild or mild to moderate type severity of COVID-19, 16.78% were moderate to worse or severe type and 0.66% had critical type of COVID-19 (*WHO, 2020*) (Table S2).

## Neuropsychiatric manifestations in long COVID-19

Neuropsychiatric symptoms were commonly reported across the studies involving long COVID-19 (LC) patients (Dataset D1). The most frequently observed manifestations included cognitive impairment, memory deficits, fatigue, depression, anxiety, sleep disturbances, and olfactory or gustatory dysfunction. These symptoms were assessed using a range of validated psychometric instruments.

Cognitive impairment and memory dysfunction were among the most consistent findings. Studies using the Montreal Cognitive Assessment (MOCA) demonstrated that LC patients frequently scored below the threshold for normal cognition, indicating both subjective and objectively measurable deficits (*Besteher et al., 2022*; *Esposito et al., 2022*; *Franke et al., 2023*; *Muccioli et al., 2023*). Other tools used included the Stroop Test (*Barnden et al., 2023*) and additional neurocognitive batteries to assess executive function, processing speed, and attention (*Díez-Cirarda et al., 2023*).

Mood disorders, including depression and anxiety, were assessed using Patient Health Questionnaire-9 (PHQ-9), Generalized Anxiety Disorder-7 (GAD-7), Depression Anxiety Stress Scale (DASS-21), Montgomery-Asberg Depression Rating Scale (MADRS), and the Hospital Anxiety and Depression Scale (HADS). These tools consistently indicated higher levels of emotional distress in LC patients compared to controls (*Besteher et al., 2022*; *Cattarinussi et al., 2022*; *Du et al., 2022*; *Muccioli et al., 2023*). Notably, some studies also reported correlations between mood symptoms and neuroimaging findings; for instance, *Cattarinussi et al. (2022)* found an association between right hippocampal connectivity and depression severity (*Cattarinussi et al., 2022*).

Fatigue and sleep-related issues were also prominent. Fatigue was assessed through the Modified Fatigue Impact Scale (*Díez-Cirarda et al., 2023*; *Muccioli et al., 2023*) and the Multidimensional Fatigue Inventory (*Cattarinussi et al., 2022*). Sleep disturbances were evaluated using tools such as the Athens Insomnia Scale (*Du et al., 2022*) and the Pittsburgh Sleep Quality Index (*Díez-Cirarda et al., 2023*), both revealing significantly poorer sleep quality in LC patients.

Post-traumatic stress symptoms (PTSS) were specifically assessed by *Tu et al. (2021)* using the PTSD Checklist for DSM-5. Their findings indicated elevated PTSS scores in LC patients, with females showing higher levels of distress and an increase in symptoms over time (*Tu et al., 2021*). Sensory dysfunctions, particularly involving smell and taste, were measured using the Brief Smell Identification Test (BSIT) and the University of Pennsylvania Smell Identification Test (UPSIT) (*Díez-Cirarda et al., 2023*; *Wingrove et al., 2023*). Persistent olfactory and gustatory issues were common and were sometimes associated with functional changes in the hippocampus and related brain regions (*Esposito et al., 2022*).

While several studies attempted to link neuropsychiatric symptoms with hippocampal structure or function, the findings were mixed. Some, like *Cattarinussi et al. (2022)*, reported positive associations, while others, such as *Besteher et al. (2022)*, did not find significant correlations. This inconsistency may stem from variations in study design, timing of assessment post-infection, or differences in patient selection criteria (Table 3 and Dataset D1).

Overall, the reviewed studies support the conclusion that neuropsychiatric symptoms are a prominent and persistent feature of long COVID-19. The use of standardized neuropsychological assessment tools lends validity to these observations, although further research is needed to better understand the underlying mechanisms and their potential associations with hippocampal alterations.

## Hippocampus in long COVID-19

The interpretation of data from neuroimaging studies and their extrapolation to AN should be tread with caution as there are various limitations to them. The variability in imaging techniques, including macrostructural, microstructural, and connectivity analyses, introduces significant assumptions and potential pitfalls (*Calamante, 2019*; *Gatto, 2020*). Specifically, variations in hardware and clinical imaging protocols across institutions pose challenges to the interpretation of diffusion MRI (dMRI) data (*Gatto,*

*2020*). Therefore, *in vivo* validation of AN requires a comprehensive approach, combining multimodal MR parameter correlations with biomarkers in cerebrospinal fluid or blood (*e.g.*, doublecortin, BDNF) (*Erickson et al., 2011*; *Just, Chevillard & Migaud, 2022*), neurocognitive assessments, and clinical parameters. While neurocognitive assessments can offer indirect insights, their reliability is limited, as cognitive changes are influenced by a complex interplay of factors beyond AN, including psychological (depression, anxiety, PTSD), pharmacological (antiepileptics, antidepressants), and lifestyle factors (sleep disturbances, substance abuse) antidepressants (*Corney et al., 2024*; *Gooden et al., 2023*; *Son & Larson, 2023*; *Su & Xiao, 2022*). Thus, none of the correlates given in the Table 1 denotes adult neurogenesis by itself. However, the current review includes findings from the studies which has any one or more of the *in vivo* correlates of AN. Findings that may help detect AN in the current review are given in Table 1.

The majority of the reviewed studies (15/17) had performed structural MRI of the hippocampus including Voxel-based morphometry (VBM), which analyzes brain images to detect regional differences in tissue composition, diffusion tensor imaging (DTI), which is utilized to visualize and quantify white matter tracts in the brain, and cerebral blood volume (CBV), which measures blood volume changes associated with brain activation (*Balsak et al., 2023*; *Besteher et al., 2022*; *Carroll et al., 2020*; *Cattarinussi et al., 2022*; *Díez-Cirarda et al., 2023*; *Ergül et al., 2022*; *Esposito et al., 2022*; *Franke et al., 2023*; *Lu et al., 2020*; *Muccioli et al., 2023*; *Rothstein, 2023*; *Taskiran-Sag et al., 2023*; *Tu et al., 2021*; *Wingrove et al., 2023*; *Yousefi-Koma et al., 2021*) while 8/17 studies had included functional MRI of the hippocampus in long COVID-19 patients (Dataset D1) (*Barnden et al., 2023*; *Cattarinussi et al., 2022*; *Díez-Cirarda et al., 2023*; *Du et al., 2022*; *Esposito et al., 2022*; *Muccioli et al., 2023*; *Tu et al., 2021*; *Wingrove et al., 2023*). Resting state fMRI was performed in seven studies (*Cattarinussi et al., 2022*; *Díez-Cirarda et al., 2023*; *Du et al., 2022*; *Esposito et al., 2022*; *Muccioli et al., 2023*; *Tu et al., 2021*; *Wingrove et al., 2023*) while *Barnden et al. (2023)* studied fMRI while the participants performed activity (Stroop task). Six studies included both structural and functional MRI study of the hippocampus in long COVID-19 patients (*Cattarinussi et al., 2022*; *Díez-Cirarda et al., 2023*; *Esposito et al., 2022*; *Muccioli et al., 2023*; *Tu et al., 2021*; *Wingrove et al., 2023*). *Franke et al. (2023)* used indirect immunofluorescence on the mouse model to study anti-neuronal and anti-glial autoantibodies in serum and CSF of PCS patients with self-reported cognitive deficits in addition to structural MRI (Table S4).

### Structural changes

Structural changes in the hippocampus varied across studies. Significant reductions in hippocampal volume were observed in post-COVID patients, particularly in hippocampal subfields such as CA1, CA3, dentate gyrus, and subiculum (*Díez-Cirarda et al., 2023*). *Carroll et al. (2020)* also reported progressive hippocampal atrophy in a case with ongoing memory deficits. Conversely, some studies documented increased hippocampal grey matter volume, which may reflect neuroinflammatory or compensatory changes (*Besteher et al., 2022*; *Lu et al., 2020*; *Tu et al., 2021*). Other studies found no statistically significant

volumetric differences between LC and control groups (*Ergül et al., 2022*; *Muccioli et al., 2023*; *Yousefi-Koma et al., 2021*) (Table 2).

### Functional connectivity

Another important *in vivo* neural correlate of adult neurogenesis by fMRI is increased functional connectivity (FC) of the hippocampus with other regions of the brain and within itself (*Burdette et al., 2010*). Again the studies included in the review presented diverse findings concerning hippocampal functional connectivity in cases of long COVID-19 varying from a comparative increase in intrinsic FC in the right hippocampus (*Cattarinussi et al., 2022*) to a lack of difference in FC of the hippocampus between LC patients and healthy controls (*Muccioli et al., 2023*). With regard to FC between the hippocampus and other parts of the brain, the studies have consensus on a reduction in FC between the hippocampus and other parts including insula in LC with olfactory disorder (*Esposito et al., 2022*), between the right head of the hippocampus and the left anterior parahippocampal division and the parietal area, including supra-marginal and postcentral areas, overlapping the dorsal attention network (*Díez-Cirarda et al., 2023*), and between the bilateral hippocampus and the whole brain in LC when compared to the control group (*Barnden et al., 2023*) (Table 2).

These discrepancies may be influenced by differences in study design, participant characteristics, symptom duration, and imaging methodology. The mixed findings across structural and functional studies emphasize the complexity of hippocampal involvement in LC and the need for further standardized investigations.

### Hippocampal perfusion

Given that increased cerebral blood flow (CBF) within the dentate gyrus of the hippocampus is an indicator of enhanced adult neurogenesis (*Steventon et al., 2020*), only two studies analysed the cerebral blood flow to the hippocampus in long COVID-19 patients which found differing results (*Díez-Cirarda et al., 2023*; *Wingrove et al., 2023*). In LC patients who were not hospitalised, vaccine naïve, and with impaired olfactory function 4–6 weeks after initial COVID-19 infection, higher CBF was observed in the posterior hippocampus when compared to those with resolved anosmia (*Wingrove et al., 2023*). While *Díez-Cirarda et al. (2023)* observed lower hippocampal perfusion in LC patients compared to healthy individuals. These contrasting findings may be due to differences in disease severity, recovery status, symptomatology, or imaging modality. While both studies support altered hippocampal perfusion in LC, further research is needed to determine whether these changes reflect compensatory neurovascular responses, inflammatory processes, or persistent injury (Table 2).

## Correlation with long COVID-19 symptoms

Several studies explored the relationship between hippocampal alterations and symptom severity in long COVID (LC) patients (Table 2). *Díez-Cirarda et al. (2023)* linked reduced hippocampal subfield volumes and perfusion with poorer cognitive performance, particularly in memory and executive functions. Similarly, *Tu et al. (2021)* found that lower left hippocampal volume was associated with higher post-traumatic stress symptoms. *Lu et*

*al. (2020)* also reported a negative correlation between bilateral hippocampal grey matter volume and memory loss.

However, findings were not consistent across all studies. *Besteher et al. (2022)* reported no association between hippocampal volume and mood or cognitive scores. *Ergül et al. (2022)* found no significant correlation between hippocampal volume and sensory dysfunction. *Taskiran-Sag et al. (2023)* noted increased hippocampal thickness in anxious patients but no correlation with inflammatory markers. These discrepancies suggest that while hippocampal changes may contribute to LC symptoms, the relationship is not yet clearly defined (Table 2 and Table S4).

### Correlation with biomarkers

Several studies explored the relationship between hippocampal alterations and biomarkers of inflammation and neurodegeneration (Table 2). *Díez-Cirarda et al. (2023)* identified elevated levels of glial fibrillary acidic protein (GFAP), myelin oligodendrocyte glycoprotein (MOG), and neurofilament light chain (NfL) in LC patients, which were significantly associated with reduced hippocampal volume and perfusion, particularly in patients who had been hospitalized. These biomarkers are linked to axonal injury, astrocyte reactivity, and myelin damage (*Hol & Pekny, 2015*; *Lorenzo et al., 2019*; *Solly et al., 1996*).

In the same study, higher levels of the chemokine eotaxin-1 (CCL11)—known to inhibit hippocampal neurogenesis—were observed in LC patients with cognitive symptoms and were negatively correlated with the volume of hippocampal subfields including the dentate gyrus, CA3 head, and CA4 head (*Díez-Cirarda et al., 2023*). These findings suggest a potential biological mechanism linking inflammation to hippocampal dysfunction and cognitive decline in LC.

However, not all studies reported such associations. *Taskiran-Sag et al. (2023)* did not find significant correlations between hippocampal thickness or volume and common inflammatory markers such as C-reactive protein (CRP) or the neutrophil-to-lymphocyte ratio (NLR). Similarly, *Du et al. (2022)* observed no relationship between hippocampal activation and systemic inflammation markers in their cohort, despite increased left hippocampal activity on fMRI.

The variability in results may reflect differences in biomarker selection, sample size, severity of acute infection, or the time elapsed since recovery. While findings from *Díez-Cirarda et al. (2023)* suggest a link between neuroinflammation and hippocampal damage in LC, the absence of consistent biomarker associations across studies indicates a need for more standardized and targeted investigations, particularly those integrating neuroimaging with biomolecular profiling.

### DISCUSSION

This scoping review aimed to summarize the existing evidence on adult neurogenesis in long COVID-19 patients with neuropsychiatric symptoms by analysing the studies on structural and functional neuroimaging of the hippocampus. Following PRISMA Extension for Scoping Reviews guidelines (*Tricco et al., 2018*), we identified 17 studies across 1,114 individuals including 687 long COVID-19 patients and 427 control subjects (*Balsak et al.,*

*2023*; *Barnden et al., 2023*; *Besteher et al., 2022*; *Carroll et al., 2020*; *Cattarinussi et al., 2022*; *Díez-Cirarda et al., 2023*; *Du et al., 2022*; *Ergül et al., 2022*; *Esposito et al., 2022*; *Franke et al., 2023*; *Lu et al., 2020*; *Muccioli et al., 2023*; *Rothstein, 2023*; *Taskiran-Sag et al., 2023*; *Tu et al., 2021*; *Wingrove et al., 2023*; *Yousefi-Koma et al., 2021*). With the obtained results, there appears to exist a noticeable dearth of research on adult neurogenesis in long COVID-19 patients. Currently, a systematic review or meta-analysis of hippocampal neurogenesis in LC is premature, owing primarily to the inadequate data, heterogeneity among the methods, variables, and population studied, and the lack of standardization across studies.

This scoping review explored the literature on hippocampal structure, function, and related molecular activity in long COVID-19 patients experiencing neuropsychiatric sequelae (Fig. 2). Overall, the findings suggest that LC is frequently associated with persistent neuropsychiatric symptoms, including cognitive impairment, fatigue, depression, anxiety, and sensory disturbances. These symptoms were commonly assessed using validated psychometric tools and were frequently accompanied by hippocampal alterations observed *via* MRI (*Barnden et al., 2023*; *Besteher et al., 2022*; *Cattarinussi et al., 2022*; *Díez-Cirarda et al., 2023*; *Du et al., 2022*; *Esposito et al., 2022*; *Lu et al., 2020*; *Rothstein, 2023*; *Tu et al., 2021*; *Yousefi-Koma et al., 2021*). While a number of studies reported volumetric reductions in hippocampal subfields and disrupted functional connectivity, findings were heterogeneous. Some studies found increased hippocampal volumes or connectivity (*Besteher et al., 2022*; *Cattarinussi et al., 2022*; *Du et al., 2022*; *Lu et al., 2020*; *Rothstein, 2023*; *Tu et al., 2021*; *Wingrove et al., 2023*), while others reported no significant differences compared to controls (*Balsak et al., 2023*; *Ergül et al., 2022*; *Franke et al., 2023*; *Muccioli et al., 2023*; *Taskiran-Sag et al., 2023*; *Yousefi-Koma et al., 2021*) or decreased hippocampal volumes or connectivity (*Barnden et al., 2023*; *Carroll et al., 2020*; *Díez-Cirarda et al., 2023*; *Esposito et al., 2022*).

Importantly, few studies specifically explored adult neurogenesis or its direct markers. Only one study assessed neurogenesis-related biomarkers—such as GFAP, NfL, MOG, and CCL11—in conjunction with hippocampal structure (*Díez-Cirarda et al., 2023*). Their findings suggest possible disruption of neurogenesis pathways, but the evidence base remains limited (*Díez-Cirarda et al., 2023*). Several other studies evaluated general inflammatory or biochemical markers (*e.g.*, CRP, NLR, LDH, vitamin B12, thyroid hormones) and examined their associations with hippocampal changes, yielding inconsistent results (*Balsak et al., 2023*; *Díez-Cirarda et al., 2023*; *Du et al., 2022*; *Ergül et al., 2022*; *Lu et al., 2020*; *Taskiran-Sag et al., 2023*).

Furthermore, although some studies found correlations between hippocampal alterations and symptom severity—particularly in relation to memory loss and post-traumatic stress symptoms (*Cattarinussi et al., 2022*; *Díez-Cirarda et al., 2023*; *Lu et al., 2020*; *Taskiran-Sag et al., 2023*; *Tu et al., 2021*; *Wingrove et al., 2023*), others reported no such associations. This variability may reflect differences in imaging methods, timing post-infection, symptom definitions, and sample characteristics.

Taken together, current evidence supports a potential link between long COVID-19, hippocampal dysfunction, and impaired neurogenesis, although the relationship remains indirect and insufficiently studied. Further research is needed to explore this connection
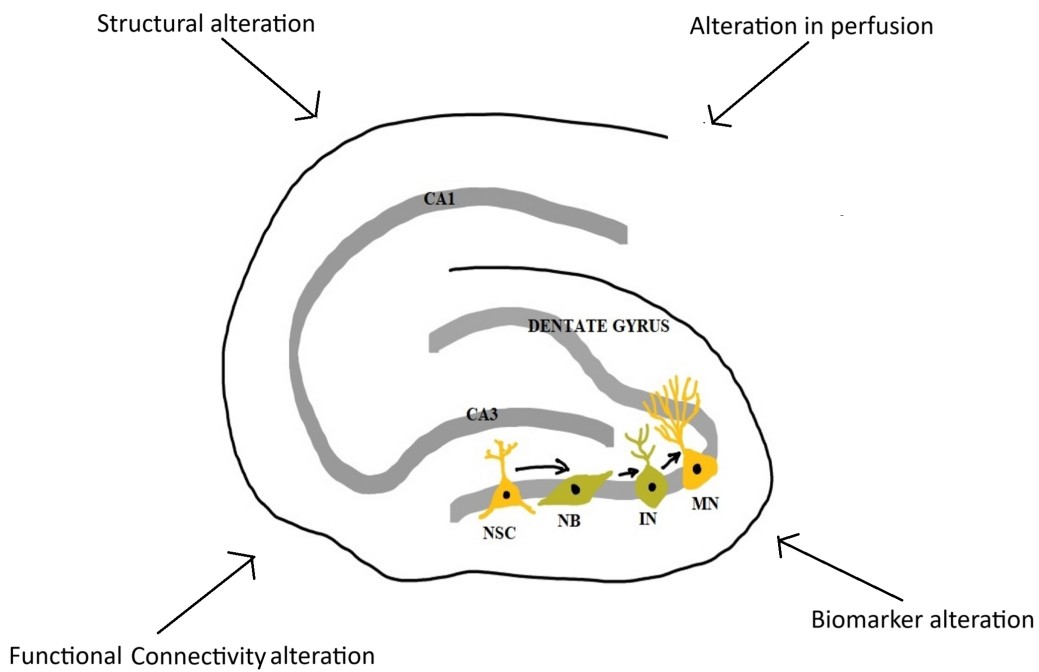

Structural alteration

Alteration in perfusion

Functional Connectivity alteration

Biomarker alteration

**Figure 2** **Major changes in hippocampus in long COVID-19.** NSC, neural stem cell; NB, neuroblast; IN, immature neuron; MN, mature neuron; CA1 and CA3, Cornu Ammonis 1 and 3 regions (Image reproduced with permission).

using longitudinal designs, neurogenesis-specific markers, and integrated neuroimaging-behavioural analyses.

## Analysis of included studies
### Methodological challenges

Significant methodological heterogeneity was evident across the literature, characterized by highly variable sample sizes and a predominant reliance on cross-sectional frameworks in study methodology.

With most of the studies being of the cross-sectional nature (*Balsak et al., 2023*; *Barnden et al., 2023*; *Besteher et al., 2022*; *Cattarinussi et al., 2022*; *Díez-Cirarda et al., 2023*; *Du et al., 2022*; *Ergül et al., 2022*; *Esposito et al., 2022*; *Lu et al., 2020*; *Muccioli et al., 2023*; *Taskiran-Sag et al., 2023*; *Tu et al., 2021*; *Wingrove et al., 2023*), there was no baseline data for comparison and no post-recuperation data to determine recovery. The included studies were single centred which could lead to selection bias, potentially resulting in participants with limited ethnic and regional diversity (*Balsak et al., 2023*; *Barnden et al., 2023*; *Besteher et al., 2022*; *Carroll et al., 2020*; *Cattarinussi et al., 2022*; *Díez-Cirarda et al., 2023*; *Du et al., 2022*; *Ergül et al., 2022*; *Esposito et al., 2022*; *Franke et al., 2023*; *Lu et al., 2020*; *Muccioli et al., 2023*; *Rothstein, 2023*; *Taskiran-Sag et al., 2023*; *Tu et al., 2021*; *Wingrove et al., 2023*; *Yousefi-Koma et al., 2021*). Moreover, the scarcity of research on potential mutants of SARS-CoV-2 across countries globally could restrict the generalizability of the study results.

There was a considerable variation in sample sizes across studies, with the majority being relatively small (median—49 subjects). Additionally, only a few studies reported effect sizes (*Besteher et al., 2022*; *Díez-Cirarda et al., 2023*; *Esposito et al., 2022*; *Franke et al., 2023*), indicating that many studies were likely underpowered. Studies with small sample sizes may yield unreliable results. In fMRI research, such small samples can lead to reduced statistical power and diminished stability in brain–behaviour correlations (*He, Tian & Lu, 2021*). Moreover, in the creation of probabilistic stimulation maps using voxel-wise statistics, small sample sizes can amplify variations in overall significance and produce less stable outcomes (*Nordin et al., 2023*; *Radua & Mataix-Cols, 2009*). Consequently, it is recommended to conduct power analysis for sample size calculation, especially in studies involving brain MRI, to ensure the reliability and reproducibility of findings.

The diverse array of tools utilized to assess neuropsychiatric parameters complicates the differentiation between investigated and yet-to-be-explored tests for hippocampal neurogenesis relevant function. There was a notable diversification in the type and method of study of neuropsychiatric signs and symptoms related to AN in the included studies. The symptoms and signs of impaired hippocampal adult neurogenesis typically include memory impairment, decline in language skills, slow decision-making, increased depression, anxieties, phobias, anger, agitation, sleep problems, disrupted circadian rhythms, and poor motor coordination (*Gope, 2020*). Additionally, adult hippocampal neurogenesis has been associated with pattern separation, forgetting, cognitive flexibility, and reversal learning (*Anacker & Hen, 2017*). Furthermore, the inhibition of adult neurogenesis due to cancer treatments has been associated with cognitive and mood-based deficits in patients (*Lim, Bang & Choi, 2018*). Whilst some of the AN related neuropsychiatric signs and symptoms in LC were studied including cognitive impairment (*Barnden et al., 2023*; *Besteher et al., 2022*; *Díez-Cirarda et al., 2023*; *Esposito et al., 2022*; *Franke et al., 2023*; *Muccioli et al., 2023*), depression (*Díez-Cirarda et al., 2023*; *Ergül et al., 2022*; *Lu et al., 2020*; *Muccioli et al., 2023*; *Taskiran-Sag et al., 2023*; *Tu et al., 2021*), anxiety (*Díez-Cirarda et al., 2023*; *Ergül et al., 2022*; *Lu et al., 2020*; *Taskiran-Sag et al., 2023*; *Tu et al., 2021*), sleep disturbances (*Díez-Cirarda et al., 2023*; *Du et al., 2022*), PTSS (*Tu et al., 2021*), stress (*Muccioli et al., 2023*), and olfactory disturbances (*Díez-Cirarda et al., 2023*; *Wingrove et al., 2023*), there is a scarcity of research for specific hippocampal neurogenesis related skills viz pattern separation, visuospatial processing. In addition, there is also a severe scarcity in the studies linking behavioural studies and biomarkers with hippocampal changes in LC. Moreover, the limited number of neuropsychological tests included in this review, coupled with significant methodological heterogeneity, hinders easy comparison of findings. Consequently, drawing robust conclusions regarding the effects of any specific outcome is challenging.

### Conceptual challenges

One of the notable limitations in the studies reviewed was the failure to link the observed changes directly to long COVID-19 and rule out the impact of other factors including the stress, depression, anxiety, and other psychological effects of social isolation and lockdown measures, comorbidities, role of medications *etc.* All but one study excluded LC patients with psychiatric illness in their study sample (*Rothstein, 2023*). Though this methodological

decision serves the purpose of eliminating the influence of existing brain MRI changes, the absence of prior history does not eliminate unreported and/or undiagnosed existing psychiatric conditions. Given that the majority of the studies being cross-sectional in nature (*Balsak et al., 2023*; *Barnden et al., 2023*; *Besteher et al., 2022*; *Cattarinussi et al., 2022*; *Díez-Cirarda et al., 2023*; *Du et al., 2022*; *Ergül et al., 2022*; *Esposito et al., 2022*; *Lu et al., 2020*; *Muccioli et al., 2023*; *Taskiran-Sag et al., 2023*; *Tu et al., 2021*; *Wingrove et al., 2023*), it becomes prudent to recommend that longitudinal studies in long COVID-19 patients are imperative to better fathom the depth of the neuropsychiatric impact of long COVID-19. Another confounder that was not studied across the studies was the role of medications in LC. Given that the study participants exhibited neuropsychiatric symptoms, any medication that they were under would have given more insight into the full extent of the impact of LC. Especially selective serotonin reuptake inhibitors (SSRIs), commonly used as antidepressants, and atypical antipsychotic medications like olanzapine and clozapine which have been linked to enhanced hippocampal neurogenesis (*Balu & Lucki, 2009*; *Boldrini et al., 2012*). The lockdown measures to curb the spread of COVID-19 have been found to be associated with increased levels of stress, with quarantine, in particular, which has been identified as a more stressful measure (*Gori, Topino & Caretti, 2022*). Additionally, factors such as anxiety levels, coping strategies, and defence mechanisms have been found to influence perceived stress during lockdowns. Drug use disorder, neurotic health symptoms, and pathological smartphone use have also been identified as contributing factors to lockdown-related stress (*Ayers, Cooper & Mayer, 2023*; *Gori, Topino & Caretti, 2022*; *Okechukwu et al., 2024*; *Paul, Mirau & Mbalawata, 2022*). Prolonged stress promotes excessive glucocorticoid levels and the hippocampus is especially vulnerable to excessive glucocorticoids given the higher levels of GC receptors (*Balu & Lucki, 2009*). Excessive GC has been associated with impaired hippocampal neurogenesis, hippocampal neurotoxicity, and dendritic atrophy (*Sapolsky, 2000*). It is worthwhile to note that there are mechanisms associated with hippocampal changes in LC patients, that occur following an injury, infection or in response to stimuli which may not be necessarily associated with alterations in AN. These include blood–brain barrier dysfunction, structural remodelling like axonal sprouting, inflammation and immune response which were not ruled out as a cause for the observed changes in the hippocampus (*Kubota, Kuroda & Sone, 2023*; *Penninx, 2021*; *Vints et al., 2024*). Thus, in addition to the methodological challenges hindering the interpretation of reported results, there are also uncertainties regarding the underlying causes of hippocampal alterations observed in LC patients. To address this unresolved question, future longitudinal studies with extended follow-up durations are warranted.

## Strengths and limitations of the review

As a planned scoping review, we adhered to established guidelines such as PRISMA-ScR (*Tricco et al., 2018*) and the Joanna Briggs Institute (*Peters et al., 2020*). However, due to the scoping review format, a risk of bias assessment was not included, hence caution should be exercised when interpreting the findings. Nevertheless, we have detected crucial methodological limitations prevalent in many included studies, offering a synthesized and

current overview of *in vivo* correlates of adult neurogenesis in long COVID-19 patients with neuropsychiatric sequelae. We detected both consistent and inconsistent findings and helped arbitrate the feasibility of a future systematic review. Additionally, as only peer-reviewed material was included, there is a possibility of overlooking relevant findings from sources not readily available (*e.g.*, conference abstracts), potentially introducing publication bias. A more extensive and rigorous systematic review would be warranted to capture relevant findings possibly missed in the current review. Furthermore, because of the limited number of studies examined and their substantial variability concerning demographic and clinical variables, their impact could not be thoroughly assessed in the current review. Methodologically, the included studies showed significant heterogeneity, with most being cross-sectional, small-scale, and single-centre. This limits causal inference, generalizability, and the ability to track recovery over time. Neuropsychiatric assessment tools varied widely, with few studies evaluating specific adult neurogenesis-related functions or conducting power analyses. Importantly, many studies lacked appropriate control groups—particularly recovered COVID-19 individuals without long-term symptoms—making it difficult to isolate the unique effects of LC from general post-infection outcomes. Conceptually, several confounding factors such as unreported psychiatric history, lockdown-related stress, comorbidities, and medication use were inadequately addressed. Furthermore, alternative mechanisms unrelated to adult neurogenesis (*e.g.*, inflammation, structural remodeling, or blood–brain barrier dysfunction) were rarely considered. These gaps hinder definitive conclusions about the specific impact of LC on hippocampal structure and function.

Nonetheless, this scoping review provides a comprehensive overview of the current literature on hippocampal alterations and neuropsychiatric symptoms in long COVID-19 (LC), identifying key methodological gaps and underexplored domains. It offers direction for future research by highlighting the need for standardized assessment tools, longitudinal designs, and integrative approaches linking behavioural, imaging, and biomarker data. Finally, as mentioned earlier in the introduction, given that direct visualisation or detection of AN is practically not possible in humans, we included articles that studied the *in vivo* correlates of AN using structural and functional neuroimaging of the hippocampus in long COVID-19 patients. Future technological improvements in the field of neuroimaging may help to overcome this challenge.

## Future directions

Future research should prioritize longitudinal studies with extended follow-up to monitor changes in adult neurogenesis in individuals with long COVID-19 and neuropsychiatric symptoms, helping to elucidate its trajectory and clinical relevance.

Mechanistic studies are also essential to determine how SARS-CoV-2 infection and its associated neuroinflammation affect AN. Evidence suggests both direct viral effects and immune-mediated responses may disrupt neurogenesis (*Ryan & Nolan, 2016*; *Song et al., 2021*), highlighting the need to identify key pathological drivers as potential therapeutic targets. Identifying biomarkers linked to altered AN could significantly enhance diagnosis and treatment monitoring. Promising candidates include brain-derived neurotrophic factor (BDNF), glial fibrillary acidic protein (GFAP), cytokines (*e.g.*, IL-6, TNF-$\alpha$),

and microglial markers (*e.g.*, CD68, Iba-1) (*Zhang & Jiao, 2015*). Peripheral or CSF biomarkers, including microRNAs and neurotrophic factors, offer accessible tools for assessing neurogenic activity (*DeKosky & Golde, 2016*). Additionally, doublecortin (DCX) and polysialylated neuronal cell adhesion molecule (PSA-NCAM) are specific to the proliferation and survival of new hippocampal neurons (*Moreno-Jiménez et al., 2021*; *Von Bohlen und Halbach, 2011*). Integrating these with neuroimaging biomarkers (Table 1) may provide a more comprehensive view of AN.

Multi-modal imaging approaches—combining fMRI, structural MRI, and PET—can further clarify the role of AN in long COVID. FLAIR MRI findings such as white matter hyperintensities (WMH), although not definitive, may be explored as neural correlates of AN (*Cho et al., 2015*; *Debette & Markus, 2010*; *Montini et al., 2021*; *Wallin & Fladby, 2010*). PET studies have revealed bilateral hypermetabolic areas in the hippocampus post-COVID-19, correlating with neurologic symptoms and severity (*Debs et al., 2023*; *Morand et al., 2022*). The possible link between brain metabolism and AN is debatable and has gained traction in the past decade. However, studies linking PET metabolic profile of the brain to adult neurogenesis are rather insufficient. Given the high metabolic demands of neurogenesis—including lipogenesis, glycolysis, and mitochondrial activity—this connection warrants deeper investigation (*Garone, De Giorgio & Carli, 2024*; *Landry & Huang, 2021*; *Rumpf, Sanal & Marzano, 2023*).

Finally, interventions that promote neurogenesis and neuroplasticity, such as serotonergic agents, atypical antipsychotics, cognitive training, or exercise, should be explored to improve outcomes in long COVID patients with neuropsychiatric symptoms. Overall, addressing these future directions could contribute to a better understanding of the role of adult neurogenesis in the pathophysiology of neuropsychiatric symptoms in long COVID-19 patients and facilitate the development of effective interventions to improve patient outcomes.

## CONCLUSION

In conclusion, this scoping review synthesized the existing evidence on hippocampal involvement in long COVID-19 patients with neuropsychiatric symptoms. To address the challenges in directly measuring adult neurogenesis *in vivo*, the included articles were selected based on their reporting of one or more structural or functional neuroimaging correlates relevant to adult neurogenesis. Despite this approach, our findings reveal a noticeable dearth of research specifically investigating adult neurogenesis in long COVID-19 patients.

The findings of this review suggest that long COVID may impact adult hippocampal neurogenesis through multiple pathways. Structural and functional alterations in the hippocampus—particularly in subfields such as the dentate gyrus and CA regions, which are critical sites of neurogenesis—were reported in several studies. Elevated levels of CCL11, a chemokine known to inhibit neurogenesis, and associations with cognitive impairment and reduced hippocampal volumes further support this hypothesis.

However, evidence remains indirect and inconsistent. Few studies directly assessed neurogenesis-specific markers or integrated behavioural and biomolecular data that

clearly distinguish impaired neurogenesis from other forms of hippocampal dysfunction. Additionally, conflicting findings regarding hippocampal volume, perfusion, and connectivity limit definitive conclusions.

Taken together, the current evidence raises important questions about disrupted hippocampal neurogenesis as a potential contributor to long COVID-related cognitive and mood symptoms. To clarify this relationship, future research should include longitudinal imaging, neurogenesis-specific biomarkers, and combined behavioural-neurobiological assessments. This review provides a foundation for future research by exploring the literature on hippocampal structure, function, and related molecular activity in long COVID-19 patients experiencing neuropsychiatric sequelae, through the lens of established *in vivo* neuroimaging correlates.

### Funding

This work was supported by the Deanship of Scientific Research, Majmaah University with research Project number R-2025-1796. The funders had no role in study design, data collection and analysis, decision to publish, or preparation of the manuscript.

### Grant Disclosures

The following grant information was disclosed by the authors:
Deanship of Scientific Research, Majmaah University: R-2025-1796.

### Competing Interests

Saikarthik Jayakumar is an academic Editor with PeerJ. Bijaya Kumar Padhi is an Academic Editor for PeerJ.

### Author Contributions

- Jayakumar Saikarthik conceived and designed the experiments, performed the experiments, analyzed the data, prepared figures and/or tables, and approved the final draft.
- Ilango Saraswathi conceived and designed the experiments, performed the experiments, analyzed the data, authored or reviewed drafts of the article, and approved the final draft.
- Bijaya Kumar Padhi conceived and designed the experiments, prepared figures and/or tables, authored or reviewed drafts of the article, and approved the final draft.
- Muhammad Aaqib Shamim analyzed the data, prepared figures and/or tables, authored or reviewed drafts of the article, and approved the final draft.
- Nasser Alzerwi performed the experiments, analyzed the data, prepared figures and/or tables, authored or reviewed drafts of the article, and approved the final draft.
- Abdulaziz Alarifi conceived and designed the experiments, performed the experiments, prepared figures and/or tables, authored or reviewed drafts of the article, and approved the final draft.

- Aravind P. Gandhi conceived and designed the experiments, performed the experiments, analyzed the data, prepared figures and/or tables, authored or reviewed drafts of the article, and approved the final draft.

## Data Availability

This is a literature review.

## Supplemental Information

Supplemental information for this article can be found online at http://dx.doi.org/10.7717/peerj.19575#supplemental-information.

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
