# Peer review of "Structural and functional neuroimaging of hippocampus to study adult neurogenesis in long COVID-19 patients with neuropsychiatric symptoms: a scoping review"

_PeerJ, doi:10.7717/peerj.19575_

## Round 0.1 · original submission · Major Revisions

Dear Authors,
The reviewers for this manuscript, suggested many points to be addressed or justified. Please incorporate or justify accordingly and resubmit asap. All the best.

Reviewer 1 ·

Basic reporting

Structural and functional neuroimaging of hippocampus to study adult neurogenesis in long COVID-19 patients with neuropsychiatric symptoms: A scoping review is an interesting and valuable review article. However, further revisions should be made in accordance with the provided recommendations to enhance its clarity and comprehensiveness

Experimental design

Methods:
• The authors should provide a rationale for why this study focuses on collecting and analyzing research articles published between 2020 and 2023.
• How was the sample size determined for the study, and is it reliable?

Validity of the findings

-

Additional comments

Discussion
• There should be suggestions to enhance the reliability of this study, aligned with previous research
• Details about each type of neuroimaging tool used should be provided to ensure better understanding for readers from other fields.

Reviewer 2 ·

Basic reporting

Language and Clarity:
- The language is often clunky and lacks precision. For instance, key technical terms are introduced without proper definitions or context (e.g., adult neurogenesis in relation to long COVID)
- Sentences are verbose and repetitive, making the manuscript difficult to read and follow. A professional copy edit would be necessary to address these issues

Introduction and Background:
- While the introduction touches on the relevance of adult neurogenesis and its potential link to long COVID, it lacks depth. Key concepts like the relationship between neuroinflammation, hippocampal function, and neurogenesis are superficially addressed
- The authors fail to justify why this review is necessary. They claim no prior reviews exist on this exact topic, yet they do not convincingly establish the novelty of their approach or its potential impact on the field

Literature Review and Referencing:
- The manuscript cites many studies but does not critically evaluate them. References are presented descriptively rather than analytically.
- Major omissions are evident, particularly studies involving PET imaging, which is a critical modality for understanding molecular and functional changes related to neurogenesis and inflammation. The absence of this perspective severely limits the scope and utility of the review.

Experimental design

Methodology:
- The review follows methodology for scoping reviews, but the execution is inconsistent and flawed.
- Search strategies are described but lack sufficient detail. The databases searched (PubMed, Web of Science, Embase, SCOPUS) are appropriate, but the inclusion and exclusion criteria are not well-justified. For instance, the decision to exclude non-peer-reviewed studies and systematic reviews is arbitrary and restricts the scope unnecessarily.

Selection of Studies:
- The final selection of studies is narrow and fails to capture the breadth of the literature. The authors provide no explanation for the exclusion of several potentially relevant studies, particularly those employing advanced imaging techniques like PET or advanced MRI methods.
- The selection bias is further exacerbated by inconsistent criteria for defining long COVID, as acknowledged by the authors themselves. This variability undermines the comparability of the included studies

Critical Analysis:
- There is no genuine critique of the methods, results, or limitations of the included studies. Instead, the authors merely summarize findings, which leads to superficial conclusions that do not advance the field.
- The review lacks synthesis. Key themes or patterns across studies are not adequately explored, leaving the reader with a fragmented understanding of the topic.

Validity of the findings

- The conclusions drawn from the review are weak and unsupported by the evidence presented. The authors make broad statements about the potential role of neurogenesis in long COVID without providing strong evidence or mechanistic insights.
- The heterogeneity of the studies included (e.g., varied methodologies, populations, and outcomes) is acknowledged but not addressed in a meaningful way. For instance, no effort is made to standardize findings or identify overarching trends.
- The manuscript fails to explore critical biomarkers and advanced imaging techniques that could provide insights into adult neurogenesis. For example, PET imaging with TSPO tracers is entirely absent from the discussion.

Additional comments

While the topic is relevant, the manuscript does not achieve its stated goal of mapping the evidence on adult neurogenesis in long COVID. The lack of depth and critical analysis renders the review largely irrelevant to researchers and clinicians

Reviewer 3 ·

Basic reporting

In this article the authors present a systematic review of the literature involving human studies assessing adult hippocampal neurogenesis or related hippocampal pathology during long-COVID. The authors use an unbiased approach to find papers related to this topic, ultimately identifying 17 primary research articles. The authors then present a highly detailed summary of each of these papers, including any changes in neuro-psychiatric symptoms, hippocampal volume, functional connectivity, and serum biomarkers of inflammation in the long-COVID patients. The authors discuss how changes in particularly hippocampal volume and cerebral blood flow may imply decreases in adult neurogenesis in long-COVID patients.

Overall, this article presents a timely review of hippocampal alterations in long-COVID patients. As research on the neurological effects of COVID-19 is a highly dynamic field with many contradictory studies, a comprehensive review on primary research in human patients is an important contribution to the literature. However, this review had several major flaws which severely dampened this reviewer’s enthusiasm:

• Adult neurogenesis itself is not explained in the introduction. This reviewer would recommend providing a paragraph describing neural stem cell differentiation, which areas of the brain are associated with neurogenesis, how adult neurogenesis is thought to contribute to cognitive processes, and it is important to ensure the reader understands that the frequency and importance of adult neurogenesis in humans is still a topic of debate.

Experimental design

Major comments:
• The authors state that hippocampal volume and blood flow are indicative of adult neurogenesis. While the authors admit that these factors are only correlated with adult neurogenesis and this reviewer appreciates the difficulty of studying adult neurogenesis in humans, it needs to be better outlined to the reader that in all of the studies identified in this review, not a single study specifically measures adult neurogenesis. The authors need to clarify that true measurement of adult neurogenesis can only reliably be done on post-mortem samples. In this reviewer’s opinion, this paper actually reviews hippocampal function during long-COVID, not adult neurogenesis, and the title/ introduction should be changed to reflect this.
• The authors do not sufficiently discuss the current evidence for decreases in adult neurogenesis in humans and/or animal models of long-COVID. The authors mention one study by Borsini et al., and briefly cite Fernandez-Castaneda et al., but do not discuss in any depth what these articles found. There is substantial evidence for decreased DCX+ neuroblasts in the subgranular zone of the dentate gyrus in humans and rodent models and this is not clear in the introduction. Please discuss the above studies more thoroughly and the following relevant studies which were not cited in the review:
o PMID: 36004663
o PMID: 38902519
o PMID: 35876935
• Throughout the results section, the authors provide a list format summary for each of the 17 papers in each subsection. This makes for difficult reading and does not synthesize the information in a useful way for the reader. I would suggest that the authors reorganize the review to provide a more succinct summary of the major phenotypes, and then simply cite the papers in which these findings are relevant. This format also makes it difficult to accurately present which results are highly supported across the different studies, and which have conflicting data.
• Table 3 is not particularly useful as it is essentially a list of the studies reviewed and their major conclusions, and does not effectively synthesize the findings. I would recommend re-structuring the table to a format such as below that places the focus on the different aspects of hippocampal pathology that were assessed across the literature.
Major Finding Volumetric hippocampal changes Funtional connectivity changes
Evidence for/against Yes/No/Conflicting Yes/No/Conflicting
Sources List citations here List citations here

• A summary figure that depicts the major changes seen in the hippocampus and outlines the pathway of adult neurogenesis for the reader would be welcome.

Validity of the findings

• In the limitations section, you should further discuss the control groups used in the papers cited. Many of these studies did not include a recovered COVID-19 control group, which is a major limitation to their analyses.

Additional comments

• Typically PASC not PACS is used as the abbreviation.
• Line 105: How can you be citing an article from 2012 as evidence of anything COVID related?
• Line 367: Define what VBM, DTI and CBV are for the reader.
• Lines 386-388: Increased hippocampal volume doesn’t reliably indicate any changes in adult neurogenesis.
• Line 593: Avoid using such strong language as “glaring”
• Lines 664: This is not an appropriate citation as both articles cited are review/opinion pieces. Please put primary literature citations.
• Lines 667- 678: Are these markers in the blood? You seem to be implying these are biomarkers, but many of these are only known to increase in the hippocampus after COVID-19, their use as a blood biomarker has no evidence. DCX particularly is not useful as a biomarker, as it is only expressed in neuroblasts in the hippocampus and/or subventricular zone.

---

## Round 0.2 · accepted · Accept

Dear authors,

I congratulate you for the acceptance of your manuscript submitted to PeerJ. This is editorial acceptance and still need to complete some pre publication tasks, so I advise you to be available for few days to avoid any delays.

All the best for your future submissions.

Reviewer 1 ·

Basic reporting

The manuscript is clearly written, well-structured, and adheres to academic conventions. Terminology is consistent, references are relevant, and the overall presentation supports comprehension of the topic.

Experimental design

The scoping review methodology is appropriate and well-articulated. Inclusion and exclusion criteria are reasonable, and the study aligns with its research objectives.

Validity of the findings

The synthesis of findings from included studies is logical and coherent. Despite inherent limitations of a scoping review, the interpretations are valid and supported by the reviewed literature.

Additional comments

The topic is timely and of importance, especially in understanding neuropsychiatric consequences of long COVID-19. The manuscript offers value to the scientific community and is suitable for publication.